# The Characteristics of Swelling Pressure for Superabsorbent Polymer and Soil Mixtures

**DOI:** 10.3390/ma13225071

**Published:** 2020-11-10

**Authors:** Jakub Misiewicz, Arkadiusz Głogowski, Krzysztof Lejcuś, Daria Marczak

**Affiliations:** 1Institute of Environmental Engineering, Wrocław University of Environmental and Life Sciences, pl. Grunwaldzki 24, 50-363 Wroclaw, Poland; jakub.misiewicz@upwr.edu.pl (J.M.); daria.marczak@upwr.edu.pl (D.M.); 2Institute of Environmental Protection and Development, Wrocław University of Environmental and Life Sciences, pl. Grunwaldzki 24, 50-363 Wroclaw, Poland; arkadiusz.glogowski@upwr.edu.pl

**Keywords:** superabsorbent polymers, SAP, swelling pressure, soil water retention, SAP-soil mixture, water scarcity

## Abstract

Superabsorbent polymers (SAPs) are used in agriculture and environmental engineering to increase soil water retention. Under such conditions, the swelling pressure of the SAP in soil affects water absorption by SAP, and soil structure. The paper presents the results of swelling pressure of three cross-linked copolymers of acrylamide and potassium acrylate mixed at the ratios of 0.3%, 0.5% and 1.0% with coarse sand and loamy sand. The highest values of swelling pressure were obtained for the 1% proportion, for coarse sand (79.53 kPa) and loamy sand (78.23 kPa). The time required to reach 90% of swelling pressure for each type of SAP differs. Samples of coarse sand mixed with SAP K2 in all concentrations reached 90% of total swelling pressure in 100 min, while the loamy sand mixtures needed only about 60 min. The results were the basis for developing a model for swelling pressure of the superabsorbent and soil mixtures, which is a fully stochastic model. The conducted research demonstrated that the course of pressure increase depends on the available pore capacity and the grain size distribution of SAPs. The obtained results and the proposed model may be applied everywhere where mixtures of SAPs and soils are used to improve plant vegetation conditions.

## 1. Introduction

Water scarcity is the main factor limiting plant survival in many regions [1,2]. The amount and quality of plant growth depends, besides the total water amount and the water use efficiency of a crop, on the relation between the water application strategy and the water holding capacity of the soil [3,4]. Therefore, the significant development of a great variety of soil amendments and smart water systems cannot be surprising [5]. Their main purpose is to retain water in the soil to be used by plants under drought stress conditions [6].

Most soil conditioners used in order to retain water are of organic origin i.e., clay minerals like kaolinite, montmorillonite, mica, bentonite, zeolite and attapulgite [7,8]. Crystalline swelling of clay minerals is mainly associated with unsaturated conditions, where water potential is controlled by partial vapor pressure, but it may also occur under saturated conditions, where the liquid water potential is controlled by high osmotic suction [9]. Another group of recently developed soil amendments are mineral composite superabsorbents that incorporate the above mentioned mineral powders to polymer structure, which, in some cases, may not only reduce manufacturing costs, but also enhance the properties (e.g., water absorbency, gel strength, and mechanical and thermal stability) of superabsorbents (SAPs) [10,11]. SAPs are one type of the maturely developed chemical water-saving agents [12,13]. SAPs are three-dimensional, insoluble, cross-linked and tissue-like polymer networks that are able to retain a large amount of water and biological fluids in their swollen state [14,15]. This can be attributed to hydrophilic residues, which are found on the polymer chains, or to counter-ions in the network, resulting in an associated osmotic pressure [16]. The cross-links resist the deformation caused by fluid absorption and stop the material from dissolving into the swelling solution [17]. Cross-linked copolymers of acrylamide and potassium acrylate undergo biodegradation in soil. SAPs are subject to solubilization, internalization, and finally incorporation or mineralization by soil microorganisms [18,19].

The prerequisite for the application of SAPs in environmental uses is to control the rate and magnitude of swelling, and understanding of gel mechanics at large deformation is paramount in ensuring the effectiveness of the products [20,21]. Compression and expansion properties of SAPs are very similar to those of clay particles, except that the magnitude of deformation of a gel bed is far larger than that of clay [22]. That is, the time for deformation is proportional to the square of the total amount of solid, and the deformation process can be described by a parabolic differential equation [22]. The driving force of swelling is the effective osmotic pressure gradient. The deformation discontinues when the swelling pressure of each gel particle is equal to the applied pressure. The liquid pressure in the gel particles is not zero even in the equilibrium compression state [22,23]. The essential factors for practical applications of SAPs, are not only absorbency, but also comprehension of the swelling rate. The water absorbency and rate of water absorption by a SAP depend on multiple factors such as pH of the swelling medium, duration of soaking, characteristics of the external solutions, such as concentration, charge number, and ionic strength, etc. [24]. Studies conducted on the swelling kinetics of various types of SAPs having different particle sizes confirm that the swelling depends on their particle size [25,26,27]. Swelling is a kinetic process that combines mass transport and mechanical deformation, which depends on the interaction between the polymer network and the solvent. We can consider swelling of SAPs or mineral composite SAPs for the layer laterally constrained by upper soil, which corresponds to practical applications of SAPs where a SAP is introduced into soil. Initially, the gel swells isotopically in the porous space of soil, at a swelling ratio. As more solvent molecules migrate into gel, the gel swells vertically while the lateral dimensions are fixed by the surrounding soil [28]. For the network of the long polymers to deform, the small molecules in the gel must change neighbors, in a process that is thermally activated, of the same kind as that in a liquid [29]. The first mode results from local rearrangement of molecules, allowing the gel to change shape but not volume. This mode of deformation occurs over a time scale that is independent of the size of the sample. The second mode results from long-range migration of the small molecules, allowing the gel to change both shape and volume [29,30]. Regardless of the manner of introduction to the soil, the SAP will swell under the load of topsoil layer and in porous spaces of soil [31]. A thorough evaluation of soil structure should take into account the geometric parameters of soil structure (size, shape, and arrangement of structural elements) along with the physical soil parameters associated to the soil structure status (bulk density, porosity, etc.) [32]. The shape and orientation of soil pores and solid phase elements allow for diagnosis of soil structure change upon external factors [33]. The swelling SAP will exert pressure on surrounding soil changing its structural integrity, and on the topsoil layer causing one-dimensional deformation [34,35]. The pore volume available for SAP is an important factor that indicates how long SAP particle can swell freely from initial, dry state, to constrained swelling that leads to exerting stress on upper soil layer [36]. The results of recently conducted research do not reflect the mechanical characteristics for swelling SAP-soil mixtures as a relationship between soil porosity and SAP particle size, which is an important parameter from the point of view of optimized and cost-effective applications in environmental engineering and agriculture.

The main objective of the present research project is to provide an analysis of the swelling pressure of SAP-soil mixtures in conditions of fixed volume and full water saturation. The authors prepared mixtures of SAPs and soils of various particle sizes and at various ratios. The objective of the study was to create a model that combines both porosity and grain size distribution of soils and SAPs to find the optimal ratio for a SAP-soil mixture and obtain the lowest swelling pressure. This results in improved stability of topsoil layer, which is desirable from the point of view of environmental engineering and agriculture.

## 2. Materials and Methods

### 2.1. Materials

Three types of cross-linked copolymer of acrylamide and potassium acrylate, Aquasorb 3005 K2 (K2), Aquasorb 3005 KM (KM) and Aquasorb 3005 KS (KS) (SNF FLOERGER, Andrézieux-Bouthéon, France), were used in this study. The SAPs used in the laboratory tests were in the form of dry, irregularly shaped granules, varied only in grain size distribution. SAP-soil mixtures were prepared by using two types of soils, coarse sand, and loamy sand in air-dry state. Laboratory tests were performed in the Mecmesin Multitest 2.5-XT (Mecmesin GmbH, Auf Rinelen, Villingen-Schwenningen, Germany) apparatus equipped with an Intelligent Loadcell (ILC) pressure sensor (measurement range 2500 N, accuracy ±0.1%), and dedicated Emperor Force™ software. (Version 1.18-408 15/10/13, Mecmesin GmbH, Auf Rinelen, Villingen-Schwenningen, Germany) For the purposes of the project, the authors developed a test script that was used to record the increase in swelling pressure in time. The authors also created laboratory equipment (Figure 1.) specially for the purposes of the test. Its aim was to maintain the conditions of full saturation of the sample during tests, while at the same time maintaining constant volume. It consisted of a Plexiglas cylinder (D = 100 mm, H = 150 mm), with watertight bottom, and with two gauges enabling the flow of water, which was kept in two containers connected to the cylinder with an elastic tube. These containers were permanently mounted on the moving arm of the apparatus in order to facilitate maintaining the water surface level accordingly to sample height. Inside the cylinder, a 3D printed porous pad was installed, of the diameter D = 100 mm (pore size D = 3 mm), that enabled water flow. It was covered with a filter paper circle (D = 110 mm, rapid filtering grade), to prevent the migration of solid particles outside the predefined sample volume. The space between the pad and cylinder walls was sealed with a rubber o-ring (D = 95 mm, d = 3 mm), placed in the indentation on the sides of the pad. Such pad is ready for placing a sample for tests. The top element of the equipment developed by the authors was the 3D printed porous pad with an indentation to mount the rubber o-ring, protected by a circular piece of filter paper (D = 100 mm). The aim of the top elements is to maintain constant sample volume and to enable venting the sample to maintain the state of full saturation throughout the duration of the experiment. The set configured in this way was placed on the base of the Mecmesin Multitest 2.5-XT apparatus and closed on the top with a steel plate (D = 70 mm) permanently attached to the ILC stress sensor.

### 2.2. Physical Characteristics of Soils and Superabsorbent Polymers (SAPs)

Grain size distribution was analysed with use of the sieving method for K2 and KM and for both soils. For each of the analysed materials, 500 g of dry sample was sieved through a set of 6 sieves with the net grid size of 0.10, 0.25, 0.50, 1.00, 2.00 and 5.00 mm. For loamy sand, which contained smaller particles, hydrometer analysis was conducted. The analysis of the KS superabsorbent was performed in the Mastersizer 2000 laser granulometer (Malvern Instruments LTD) (Malvern Panalytical Ltd, Malvern, United Kingdom) with SCIROCCO 2000 module. Measurements were taken on dry aerosol, at the pressure of 2 bars with the measured particle diameter range of from 0.020 to 2000 μm. Each test was conducted 3 times, and the results were then averaged. Soil bulk densities were determined by the bulk density test, in which the weight of taken soil sample volume was measured after they had been dried in the oven at 105 °C for 24 h to obtain constant mass. Soil bulk densities (1) and porosities (2) were calculated using the following equations:(1)$$\rho_{d\;=\;\frac{m_{s}}{V}\left[\frac{g}{{\mathrm{cm}}^{3}}\right]}$$
(2)$$e=\frac{\rho_{s}-{\;\rho }_{d}}{\rho_{s}}\left[-\right]$$
where ρ_d_ is soil bulk density, ρ_s_ is soil specific weight, V is soil sample volume and m_s_ is soil dry mass.

Soils and SAPs parameters are presented in Table 1. Grain size distribution curves with range of grains were created (Figure 1). The tested soils were classified according to USDA (United States Department of Agriculture and the National Cooperative Soil Survey) classification.

### 2.3. Swelling Pressure Measurement Procedure

Samples were prepared for each test in form of a mixture of SAP and soil in dry-aerated condition, at the set proportions (0.3%; 0.5%; 1.0%) and the total weight of 100 g. The mixture was then mixed manually in the container until homogenous mixture was obtained. Samples prepared in this way were placed in the measurement cylinder, on the porous pad covered with filtration paper and sealed with the o-ring. Before starting the test, the sample was levelled to the predefined level. Then the whole sample was covered with a circular piece of filter paper and porous top ring (D = 100 mm) with an o-ring, so as to maintain constant volume of the sample during the test and to avoid friction against the cylinder walls, which might affect the readings of the sensor. The measurement of swelling pressure began at the moment when the measurement cylinder was filled with deionized (DI) water so as to achieve the state of full saturation of the sample. The increase of pressure in time measured by the ILC sensor was recorded at the frequency of 10 Hz and presented in form of a diagram of the relation between swelling pressure and time. The swelling pressure measurement lasted for 240 min for each sample. The tests were conducted at constant ambient temperature of 20 °C.

### 2.4. Model Estimation (Curve Fitting)

The model estimation consisted of 27 iterations of the experiment for each of the soils, i.e., coarse sand and loamy sand (54 experiments). For three different SAP-soil proportions and three different grain sized SAPs (KS, KM, K2), three identical repeatable experiments were conducted, where the pressure was measured with 10 s interval. The main aim was to normalize experimental data by extracting the mean characteristic of each SAP (e.g., concentration, expansion) and soil to make all experiments independent and comparable [37,38]. This procedure allowed us to examine all experiments and to attempt to create one universal formula to predict the behavior of soil mixture with SAP, assuming that it depends on known SAP concentration, grain size and expansion characteristics as well as on such soil properties as porosity and grain size distribution. The calculations and curve fitting were conducted in R environment with use of the pracma package, that was designed to apply numerical mathematical functions in various fields of science, including environmental and engineering sciences [39].

### 2.5. A Model for Swelling Pressure

To obtain independent and comparable samples we assume the linear assessment of swelling pressure in sample caused by SAP concentration (e.g., 0.3%, 0.5% and 1.0%) and extract the influence of different concentrations. The grain size influence of SAP was extracted by the maximum value from nine experiments for one superabsorbent size after scaled concentration (Figure 2). The type of SAP was used as the identification code for each trajectory, followed by sample number and the proportion of SAP in the mixture with soil. Such identification was applied both to loamy sand and coarse sand. (e.g., K2_1_03 refers to K2 SAP, first sample at the concentration of 0.3% in the soil mixture). All normalized trajectories were the basis for calculating the mean course of SAP swelling pressure and the standard deviation (σ^1^) for the analysed process according to the so-called “three-sigma rule of thumb” to describe about 70% of empirical data [40].

The obtained mean course of SAP expansion has similar properties to the cumulative distribution function resulting from Equation (3):(3)$$F\left(x\right)=1-e-\lambda x$$
where: λ is a slope coefficient correlated with porosity Coe_pr_ [-] and superabsorbent expandability Coe_sup_ and it can be described by Equation (4):(4)$$\lambda =Coe_{pr}\;\times \;Coe_{sup}$$

After the modification to describe the pressure of the superabsorbent/soil mixture, which is a fully stochastic model and can be read as:(5)$$F\left(x\right)\;=\;0.8\;\times \;\left(1-e^{-Coe_{pr}\;\times \;Coe_{sup\;}\times \;x}\;\right)\;\times \;Conc_{sup}\;\times \;Coe_{sup}$$
where: Coe_sup_ is the coefficient of superabsorbent swelling pressure [kPa], Conc_sup_ is the concentration of superabsorbent mixed with soil [%], and x is the time passed after the first contact with water [min].

Equation (3) described in this way reflected the course of the analysed process in the way presented in Figure 3. The course of the model and mean trajectories was similar to the theoretical expansion rate for polymer hydrogels obtained by Iwata et. al. [22].

The obtained modified cumulative distribution function was verified for all SAP concentrations and grain size distributions that were used in the research project. The authors found that the coefficients described above are strongly dependent on soil properties.

## 3. Results

### 3.1. Physical Characteristics of Soils and SAPs

The grain size distribution both of SAPs and of the tested soils was analysed in order to determine the correct methods for describing the swelling pressure of superabsorbents in a porous medium. Not only is the morphology of SAP particles a critical factor, but the particle size and swelling capacity are also important aspects in the consideration of when these particles are used in various applications [41,42]. Another parameter that is important from the point of view of SAPs and soil mixtures is soil porosity (e) which indicates the available volume. Basic physical properties of the used soils and SAPs are presented in Table 1. Grain size distributions of the tested SAPs and soils are presented in Figure 4.

For coarse sand, particles from the 1.00–2.00 mm range account for the highest share (60.0%), followed by 0.50–1.00 mm range with 35.0% share. Loamy sand was characterized by the most differentiated particle sizes. The ranges of 0.10–0.25 mm, 0.25–0.50 mm and 0.50–1.00 mm accounted for 32.0%, 27.0% and 17.0%, respectively. SAP KM was characterized by the most homogenous structure with 74.0% of particles from range 0.50–1.00 mm. The second SAP, K2 was characterized by ranges of grain sizes 0.25–0.500 mm, 0.50–1.00 mm, 1.00–2.00 mm and 2.00–5.00 mm accounted for 2.0%, 17.0%, 61.0% and 20.0%, respectively. For KS, the highest share was noted for the 0.10–0.25 mm particles (50.0%), followed by 0.25–0.50 mm and 0.05–0.10 mm accounted for 30.0% and 14.0%, respectively.

### 3.2. Results and Model Verification for Coarse Sand and Loamy Sand

The results demonstrate that the percentage of SAP in the mixture has a noticeable influence on the increase in swelling pressure of SAP-soil mixtures. Regardless of the soil used, the values of swelling pressure are very similar for the same SAP-soil proportions for K2 and KM (Figure 5 and Figure 6). SAP KS is susceptible to the influence on the relationship between SAP grain size and soil porosity (Figure 5 and Figure 6). The highest values of swelling pressure were obtained for the 1% proportion K2, for coarse sand (79.53 kPa) and loamy sand (78.23 kPa). In comparison, the samples of soil- KM mixtures (1%) for coarse sand and loamy sand reached maximum swelling pressure values of 62.22 kPa and 65.67 kPa, respectively. For the same proportion of 1% in the KS mixed with coarse sand and loamy sand the biggest differences in swelling pressure values occur. In this variant, the maximum swelling pressure values obtained were 37.16 kPa for coarse sand and 53.15 kPa for loamy sand. The difference in maximum swelling pressure in both types of used soils for K2 between 1.0% and 0.50% concentration was 1.60%, and between 0.50% and 0.30% concentration 3.30%. Similar differences between coarse sand and loamy sand were observed for KS, but the values were slightly higher, 4.20% between 1.00% and 0.50% and 8.20% between 0.50% and 0.30% concentrations. The mixtures of KM with coarse sand and loamy sand showed an opposite trend and the highest pressure values with swelling pressure results with differences of 10.00% between concentrations 1.00% and 0.50%, and 8.00% between concentrations 0.50% and 0.30%. The greatest differences between both mixtures resulted from the time required to reach 90.00% of swelling pressure for each type of SAP. Samples of coarse sand mixed with K2 in all concentrations reached 90% of total swelling pressure in 100 min, while the loamy sand mixtures needed only about 60 min. KM mixed with loamy sand also showed uniform values of time required to reach 90.00% of maximum swelling pressure among all tested SAP-soil proportions but in this case it took only about 18 min. KM mixed with coarse sand showed more variety between tested concentrations with 25.50 min, 32.83 min, and 43.83 min needed to reach the taken values. The shortest time needed to reach 90% of total swelling pressure was noted for mixtures of loamy sand and KS, which was 8.83 min, 17.33 min, and 10.17 min for the proportions 0.30%, 0.50% and 1.00%, accordingly. The variations of time needed for the sample to reach 90% of maximum value of pressure were the greatest for coarse sand and KS, and they were 29.00 min, 134.00 min and 174.00 min for 0.30%, 0.50% and 1.00% respectively. The values of maximum swelling pressure and time to reach 90% of maximum swelling pressure for analyzed samples are presented in Table 2.

The equation presented in Section 2.5 was used to verify the measured expansion of SAP in loamy sand (Figure 5) and coarse sand (Figure 6). For low concentrations (0.30% and 0.50%) the model gave satisfying results both for types of soil and for three different grain sized SAPs. The estimated courses of expandability in form of the proposed equation were in the range of variability that was measured for each presented scenario. The SAP swelling pressure was different for the same grain size in different soils. This was related to porosity properties of the soil, different SAP grain size, change of apparent density and voids content of the mixture [43].

However, the experiment on KS and loamy sand for high concentration revealed an entirely different expandability dynamics. The 1% concentration of KS in coarse sand still continued to expand after 4 h, while the form of all other concentrations and SAP grain sizes stabilized after 10 min (loamy sand KS 1%) to 1.5 h (coarse sand K2 1%). This process may even continue for about 4 following days. The reason of this is the high concentration of small size SAP particles (0.1 mm) and high porosity of coarse sand. High expandability of small-size SAP exerts stress on the structures of porous soil affecting the stable structure and pushing it into the empty spaces in the soil sample.

## 4. Discussion

It is difficult to record the nature of SAP swelling experimentally, especially in the initial, most dynamic phase of transient swelling [17]. As far as SAP-soil mixtures are concerned, this phase has a significant influence on the final swelling pressure. This phenomenon is caused by the ratio of soil pore volume to the sizes of SAP particles. This is particularly noticeable for the KS superabsorbent (grain size 0.10–0.50 mm) in the mixture with coarse sand (average pore size = 0.1 mm [44]), where the nature of the increase in swelling pressure is different to that of other samples and seems less adjusted to the model. This results from the fact that the pore volume is comparable to the size of a single SAP particle when the SAP is in a dry state and the relatively high degree of swelling of SAP particles when they fill the available volume. At soil porosity of 32% and 0.5% or 1.0% share of the KS SAP, the state of swelling pressure balance emerges after a very long time. SAPs consisting of large particles, i.e., K2 (1.00–2.00 mm) and KM (0.50–1.00 mm) will reach similar values of swelling pressure for the given proportion regardless of the type of soil in which they are placed, because due to their size, they have limited possibilities to expand in the available pore capacity.

The application of the calculation-based approach for SAP-soil mixtures, such as for expansive soils [38] for 2-phase media, is inappropriate. The SAP-soil mixture in the full saturation state is a 3-phase medium, where the share of SAP influences the mechanical properties of the mixture [12]. This is noticeable while testing absorbency under load (AUL), where the higher amount of SAP in the mixture results in higher swelling pressure, which results in pushing out the top layer of soil [35,45]. Due to that, the nature of swelling of SAP-soil mixtures requires special attention, especially from the point of view of practical applications, where the SAP is applied in superficial layers of soil.

Optimised application of SAPs should be based on choosing such grain size distribution of SAPs that will be adequate to the type of substrate and selecting a proper dose. A lower share of SAP always results in lower swelling pressure, and thus it will have a smaller influence on pushing out the top soil layers. Additionally, literature provides examples of cases when a lower share of SAP in the mixture led to higher efficiency of its application [46,47,48]. Lower doses of SAP (0.30%, 0.50%) will be more adequate for practical applications in environmental engineering and agriculture due to their lower influence on the top layer of soil and higher AUL values [35,49]. Both problems may be solved by using a technology that will allow for free water absorption by SAP in spite of the soil load and that will eliminate the problems resulting from mixing SAPs directly with soil [50,51,52].

## 5. Conclusions

During swelling, the SAP-soil mixture exerts pressure on the top layer of soil, and the pressure is directly related to the dose of SAP in the mixture. The course of pressure increase depends on the available pore capacity and the grain size distribution of SAPs. These two factors also influence the time necessary to reach the maximum value of swelling pressure. SAPs consisting of large particles (K2-2.0–5.0 mm, KM-0.5–1.0 mm) exert similar swelling pressure, regardless of the porosity of soil in which it is placed. On the other hand, SAPs with finer grains (KS-0.1–0.5 mm) placed in a medium of the porosity of 32% will exert only slight stress (about 20 kPa) on the top layer of soil in the initial phase of swelling, but the pressure will increase gradually to achieve stability after a significantly longer time. At the porosity of 23%, fine-grained SAP (KS-0.1–0.5 mm) will achieve the maximum swelling pressure in a short time, about 9 min. However, the same SAP surrounded by a highly porous medium (32%) will require much more time to achieve the state of maximum pressure and balance: 240 min and more. The obtained results and the proposed model may be applied everywhere where mixtures of SAPs and soils are used to improve plant vegetation conditions.

## Figures and Tables

**Figure 1 materials-13-05071-f001:**
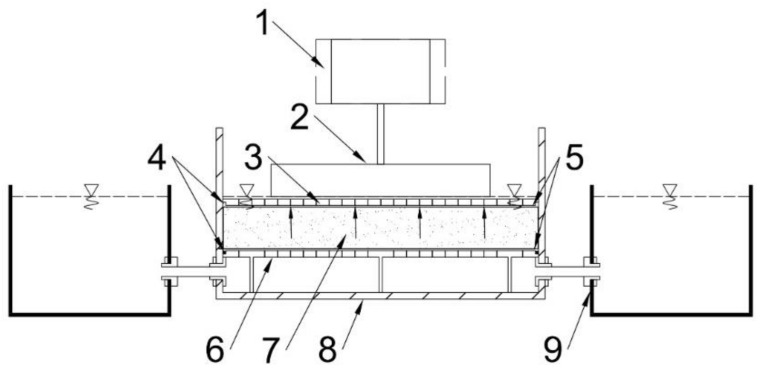
Test apparatus (Mecmesin multitest-2.5xt) with the experimental setup. (**1**) ILC pressure sensor, (**2**) pressure plate, (**3**) porous upper plate, (**4**) rubber o-ring, (**5**) filter paper, (**6**) porous base plate, (**7**) superabsorbent polymer (SAP)-soil mixture, (**8**) experimental cylinder, (**9**) deionized (DI) water reservoir.

**Figure 2 materials-13-05071-f002:**
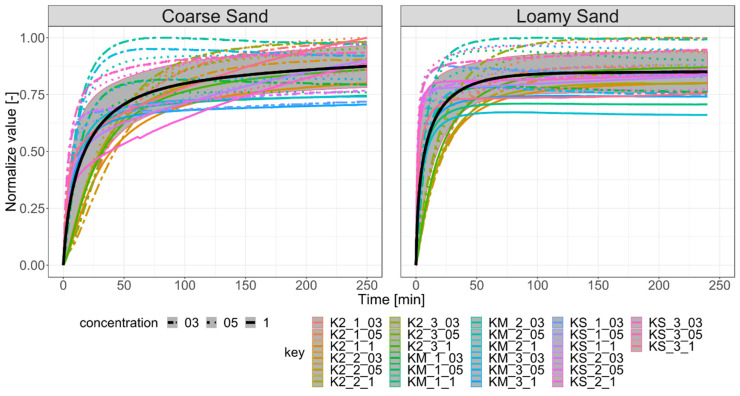
Normalized trajectories of experiments for coarse sand (**left**) and loamy sand (**right**) with the mean course and σ^1^ range of acceptance.

**Figure 3 materials-13-05071-f003:**
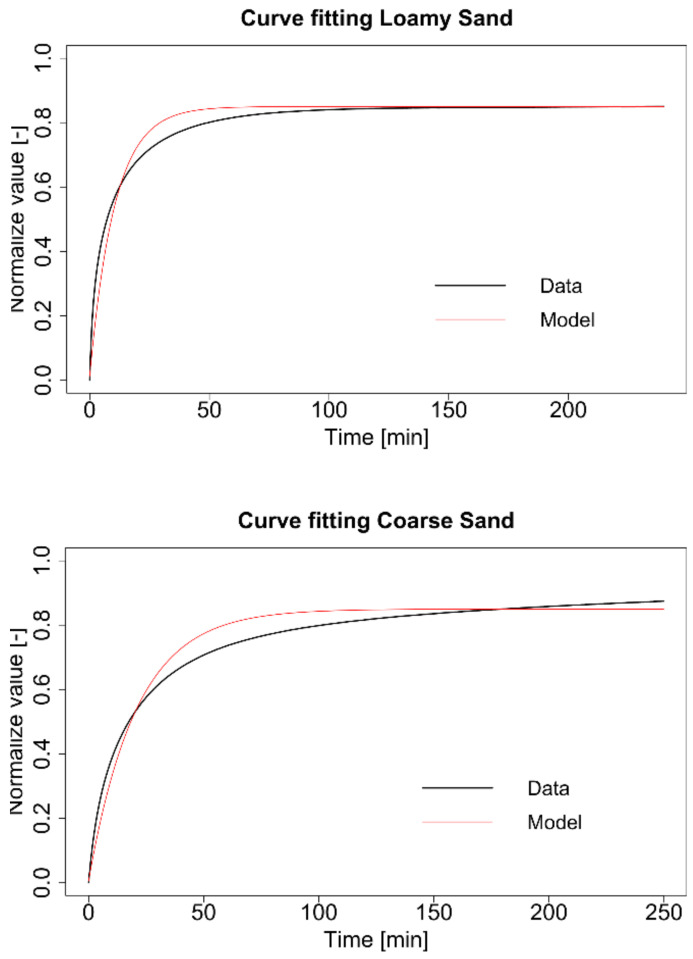
Fitting mean empirical course (black line) with the modified equation of cumulative distribution function (red line) for loamy sand (**up**) and coarse sand (**down**).

**Figure 4 materials-13-05071-f004:**
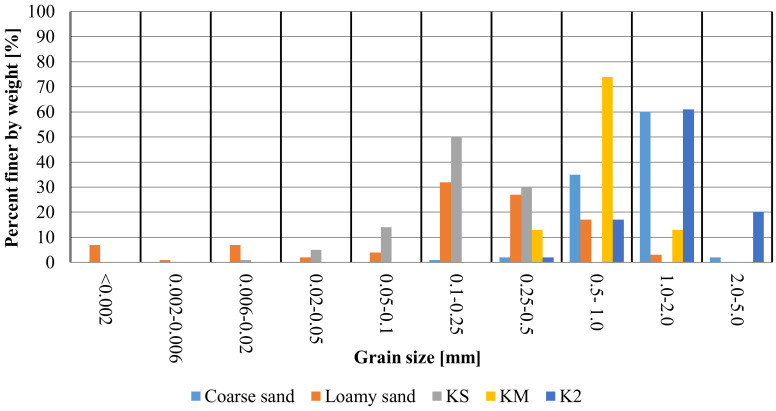
Grain size distributions of the tested SAP and soil.

**Figure 5 materials-13-05071-f005:**
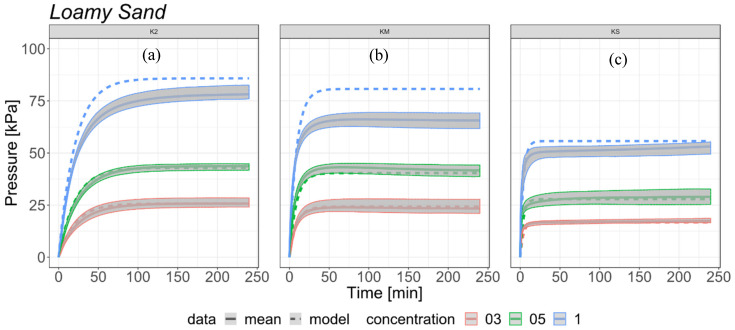
Comparison of the estimated model (dotted line) with measured data (continuous line) of SAP-loamy sand mixtures with range of variability (grey area) based on all experiments in the selected scenario: (**a**) K2; (**b**) KM; (**c**) KS.

**Figure 6 materials-13-05071-f006:**
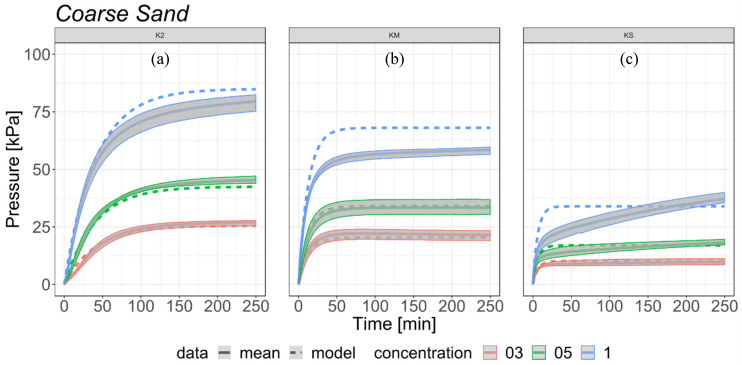
Comparison of the estimated model (dotted line) with measured data (continuous line) of SAP-coarse sand mixtures with range of variability (grey area) based on all experiments in the selected scenario: (**a**) K2; (**b**) KM; (**c**) KS.

**Table 1 materials-13-05071-t001:** Basic physical properties of the used soils and SAPs.

Physical Properties		Soil/SAP				
		Coarse Sand	Loamy Sand	K2	KM	KS
Specific Weight [g/cm^3^]		2.65	2.65	1.10	1.10	1.10
Bulk Density [g/cm^3^]		1.8	2.05	NA	NA	NA
Porosity [-]		0.32	0.23	NA	NA	NA
Grain Size Distribution [%]	Gravel (>2.00 mm)	2	0	20	0	0
	Sand (0.05 mm–2.00 mm)	98	83	80	100	94
	Silt (0.002 mm–0.05 mm)	0	10	0	0	6
	Clay (<0.002 mm)	0	7	0	0	0

NA = Not Applicable/Negligible.

**Table 2 materials-13-05071-t002:** Average maximum swelling pressure for tested samples.

Soil	SAP	SAP-Soil Proportion [%]	MAX Swelling Pressure [kPa]	Time-90% Max Pressure [min]	Standard Deviation of Sample [kPa]
Coarse Sand	K2	0.30	27.00	103.00	0.250
		0.50	45.37	100.00	0.235
		1.00	79.53	106.50	0.195
	KM	0.30	21.49	25.50	0.158
		0.50	33.35	32.83	0.146
		1.00	62.22	43.83	0.107
	KS	0.30	9.67	29.00	0.119
		0.50	18.68	134.00	0.133
		1.00	37.16	174.00	0.162
Loamy Sand	K2	0.30	25.71	60.50	0.194
		0.50	43.28	56.67	0.186
		1.00	78.23	64.00	0.162
	KM	0.30	23.39	18.83	0.148
		0.50	41.63	16.83	0.118
		1.00	65.67	18.15	0.083
	KS	0.30	17.39	8.83	0.070
		0.50	28.98	17.33	0.102
		1.00	53.15	10.17	0.067

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
