# Peer review of "The Characteristics of Swelling Pressure for Superabsorbent Polymer and Soil Mixtures"

_materials, 2020, doi:10.3390/ma13225071_

Round 1

Reviewer 1 Report

The manuscript "The Characteristics of Swelling Pressure for Superabsorbent Polymer and soil mixtures" shows the usuage of Aquasorb commercial available based on polacrylamide polymers cross linked with potassium acrylate  as superabsorbent polymers and their influence  in different mixture ratio on course and loamy sand. So far the results interesting and there only minor points to address.

As for a general question if hydrogels based on polacrylamide applied how are the long term biodegradable properties of such if exposed in nature? Maybe it would give some values to the manuscript that such hydrogels are no threat to enviromental life.

Using different ratios of 0.3, 0.5 and 1% mixing with coarse and loamy sand why not higher amount than 1% or lower than 0.3%?  The authors shown that the water uptake  was very fast but how fast is the release, are some rest water left in SAP after a period of time?

If such particles applied, by looking in practical applications how long can they retain in soil or ground before become ineffective. Using different pressure how does it reflect to nature enviroments, did the authors verify this?

Some minor points

Page 12 line 315. The authors said "Additionally in literature....". Please add references

Reviewer 2 Report

Review comments on

materials-985383-peer-review-v2

The Characteristics of Swelling Pressure for
Superabsorbent Polymer and soil mixtures

The paper deals with the results of swelling pressure of three cross-linked copolymers of acrylamide and potassium acrylate mixed at the ratios of 0.3%, 0.5% and 1.0% with coarse sand and loamy sand. The authors demonstrated that the course of pressure increase depends on the available pore capacity and the grain size distribution of SAPs. The manuscript contains mistakes.

Detailed comments are given below and it must be revised based on these,

  1. A space shall be added before every reference bracket.
  2. 3 is unclear, missing information and shall be checked and improved.
  3. 4 is unclear, missing information and shall be checked and improved.
  4. 6 is unclear, missing information and shall be checked and improved.
  5. Why Fig. 2 appears after Fig. 4, this is incorrect. Please revise.
  6. Line 270: Tab. 2? It is wrongly typed.
  7. Line 51: “A thorough evaluation of soil structure should take into account the geometric parameters of soil structure (size, shape, and arrangement of structural elements) along with the physical soil parameters related to the soil structure status (bulk density, porosity, etc.).”, a reference shall be made here, e.g., Chen, J.J., Ng, P.L., Chu, S., Guan, G.X. and Kwan, A.K.H., 2020. Ternary blending with metakaolin and silica fume to improve packing density and performance of binder paste. Construction and Building Materials, 252.
  8. Lines 275-278: “The estimated courses of expandability in form of the proposed equation were in the range of variability that was measured for each presented scenario. The SAP swelling pressure was different for the same grain size in different soils. This was related to porosity properties of the soil and different SAP grain size.” This is actually wet packing method, the explanation shall be made with references, e.g., Lejcuś, Krzysztof, Michał Śpitalniak, and Jolanta Dąbrowska. "Swelling behaviour of superabsorbent polymers for soil amendment under different loads." Polymers 10.3 (2018): 271.

-END-

Author Response

The authors would like to thank Reviewer 3 for the thorough review of our manuscript and for providing us with their comments and suggestions on how to improve the quality and clarity of the manuscript.

Please find a list of all the answers to the Reviewer 3 comments below.

Reviewer #3 wrote: 

  1. A space shall be added before every reference bracket.

Our response:

The comment has been taken into account. A space was added before each reference bracket.

Reviewer #3 wrote: 

2.3 is unclear, missing information and shall be checked and improved..

Our response:

The comment has been taken into account. Section 3. has been checked and corrected.

Reviewer #3 wrote: 

3.4 is unclear, missing information and shall be checked and improved.

Our response:
The comment has been taken into account. Section 4. has been checked and slightly corrected.

Reviewer #3 wrote: 

4.6 is unclear, missing information and shall be checked and improved.

Our response:

There is no such point in the text. Section 5. has been checked and slightly corrected.

Reviewer #3 wrote: 

5.Why Fig. 2 appears after Fig. 4, this is incorrect. Please revise.

Our response:

The comment has been taken into account. The numbering of figures and tables has been revised.

Reviewer #3 wrote: 

6.Line 270: Tab. 2? It is wrongly typed.

Our response:

The comment has been taken into account. (line 270)

Reviewer #3 wrote: 

7.Line 51: “A thorough evaluation of soil structure should take into
account the geometric parameters of soil structure (size, shape, and
arrangement of structural elements) along with the physical soil
parameters related to the soil structure status (bulk density, porosity,
etc.).”, a reference shall be made here, e.g., Chen, J.J., Ng, P.L.,
Chu, S., Guan, G.X. and Kwan, A.K.H., 2020. Ternary blending with
metakaolin and silica fume to improve packing density and performance of
binder paste. Construction and Building Materials, 252.

Our response:

The comment has been taken into account. (Now line 85)

Reviewer #3 wrote: 

8.Lines 275-278: “The estimated courses of expandability in form of the
proposed equation were in the range of variability that was measured for
each presented scenario. The SAP swelling pressure was different for the
same grain size in different soils. This was related to porosity
properties of the soil and different SAP grain size.” This is actually
wet packing method, the explanation shall be made with references, e.g.,
Lejcuś, Krzysztof, Michał Śpitalniak, and Jolanta Dąbrowska. "Swelling
behaviour of superabsorbent polymers for soil amendment under different
loads." Polymers 10.3 (2018): 271.

Our response:

The comment has been taken into account. (Now line 288)

Reviewer 3 Report

Dear editor, dear authors,

The present paper is certainly innovative because it presents an interesting lab-made method to measure the swelling pressure of a soil added of superabsorbent polymers.

Even if the main findings are not that surprising, meaning that the more SAP is added to the soil and the bigger is their granulometry, the higher is the pressure observed, I am favorable for publishing.

However, I would like to propose a couple of suggestions to further improve the paper quality:

  1. Please highlight in the experimental section that the only difference between K2, KS and KM is their granulometry. Report them in table 1 and 2 with the same order, so as for fig.5 and 6 (so change KS with K2 in tab.1)
  2. Figure 2 will be easier to understand with bars and not a sum of data (Authors reported the sum of smaller in every point: e.g. K2 at 2mm result 80%, indeed is 3% 0,5 mm, ca 17%1 mm and 60% 2 mm). Better would be see the distribution (Bell-shaped like, even if it will not be so).

Minor typos:

  • Figure 2, y-axys= percent
  • First 2 sentences of the conclusions should be joined with “and”

Otherwise, the paper is otherwise well written, clear and understandable.
